# Tests of Belt Linear Speed for Identification of Frictional Contact Phenomena

**DOI:** 10.3390/s20205816

**Published:** 2020-10-14

**Authors:** Piotr Bortnowski, Lech Gladysiewicz, Robert Krol, Maksymilian Ozdoba

**Affiliations:** Department of Mining and Geodesy, Faculty of Geoengineering, Mining and Geology, Wroclaw University of Science and Technology, Na Grobli 15, 50-421 Wroclaw, Poland; piotr.bortnowski@pwr.edu.pl (P.B.); lech.gladysiewicz@pwr.edu.pl (L.G.); robert.krol@pwr.edu.pl (R.K.)

**Keywords:** belt conveyor, linear belt speed, elastic slippage

## Abstract

In the locations where driving forces are transmitted, the changing tensile forces cause rapid elastic deformations of the belt. The deformation changes the belt speed. Measurement of the belt speed on the friction contact sections is essential to identify elastic slippage. However, the scale of the phenomenon is small, so it is necessary to use precise measuring equipment. The article presents measurements of the linear belt speed with the use of various sensors and measuring devices. A measurement error was determined for each of the presented measurement methods. The method with the highest accuracy was used to identify the elastic slippage on the drive pulley.

## 1. Introduction

In conventional belt conveyors, the belt is driven by the drive pulleys located in the head station. Occasionally, drive pulleys are also installed in the tail station, or along the conveyor flight, in intermediate pulley or belt drives. In every type of drive mechanism, the belt receives the driving force owing to the frictional contact phenomenon. The transmission of driving forces results in rapid changes of the tensile forces acting on the core of the belt. As the driving forces balance the resistances to motion, the changes of forces in the frictional contact lengths (in the drive mechanisms) are equal to the changes of forces in the belt along the conveyor route. Tension changes on friction contact areas such as pulleys or intermediate drives are significantly faster than on the flight (along the conveyor route). In the locations where driving forces are transmitted, the changing tensile forces cause rapid elastic deformations of the belt. These deformations are accompanied by the phenomenon known as elastic slippage. The elastic slippage of the belt on the drive unit is understood as the relative movement of belt to the surface of the drive pulley without loss of friction contact. The phenomena are caused by elastic deformation of the textile core of belt, as a result of tensile force changing. It occurs when the conditions of frictional contact work stably. Elastic slippage is accompanied by the shear deformation of the rubber surfaces of belt and surface of drive pulley. In the other side, sliding slippage is the displacement of belt to the surface of drive pulley, known as a result of traction loss [1]. If the belt is tensioned properly in order to meet the frictional contact condition, only elastic slippage is observed. Exceeding the limit friction conditions results in an additional sliding slippage. The driving energy is always transmitted from the driving element (a pulley or a driving belt) of a higher speed to the band of a lower speed (the driven belt). The slippage causes frictional wear of the belt bottom cover and of the pulley lining. Generally, the slippage on the pulley drive or belt drive mechanisms is the sum of the elastic slippage and the sliding slippage. In the case of an intermediate belt drive, tests of the impact of slippage are crucial, as slip determines the distribution of loads on individual conveyor drives (head drives or intermediate drives). Slippage tests require measuring devices of high accuracy, as slippage speeds remain on the order of several tenths of millimeters per second. Standard devices for measuring belt linear speed do not meet such requirements. Therefore, research is performed into new measurement devices and techniques which could prove useful in testing the influence of elastic slippage on the operation of intermediate drives. The search for high-precision linear belt speed measurement methods includes laboratory testing. It should be noted that these conditions are different from the actual conditions prevailing in mines and industry. The effectiveness of the measurement will be influenced by negative factors. Humidity, dust, vibrations, and a number of other disturbances will affect the accuracy and usability of the selected methods. In order to learn about the issue and to develop empirical measurement methods, laboratory tests are necessary, even under simplified conditions.

The paper is organized as follows: In Section 2, we describe the known methods of measuring linear speed of a conveyor belt, with regard to their approximate level of accuracy. In Section 3, we present the adopted procedure of investigation on the intermediate drive test rig that keeps stabilized laboratory conditions for the three preselected measuring methods. Furthermore, we describe the method of implementation of the chosen methods to the actual conditions of testing in the laboratory. In Section 4, we evaluate the accuracy of the proposed three testing methods with the standard measures, i.e., mean absolute error (MAE) and root mean square error (RMSE). In Section 5 we present the example of belt speed measurement on the drive pulley that proves the usefulness of the optical method based on a high-speed camera in tests of elastic slippage in the frictional contact section. The last section concludes the paper. 

## 2. Procedures

### 2.1. Laboratory Facility

Tests of the intermediate belt drive mechanism are currently performed in the Mining Machinery Systems Laboratory at the Department of Mining and Geodesy, Wroclaw University of Science and Technology [2]. They required constructing a dedicated test rig (Figure 1), comprising two flat belts 400 mm in width: the first belt is the driven belt and the second belt is driving belt of the intermediate drive. The intermediate drive consists of a belt loop 12 m in length, which is together with the pulleys installed in the main conveyor so as to provide a section of frictional contact 5 m in length. The rig is equipped with three drives: the head drive and the tail drive of the main conveyor, and the head drive of the short inner conveyor functioning as the intermediate drive. 

The tests are among others aimed at determining the relationship between the circumferential force applied on the intermediate drive as a function of belt linear speed. This speed may be calculated from the instantaneous revolutions of the intermediate drive motor. During preliminary tests, the circumferential force on the intermediate drive showed great variability (Figure 2). This variability of the values measured at constant parameters (in this case at constant motor revolutions) is typical for experimental tests and may be caused by a number of factors, such as the implemented measurement system, the influence of both the driven belt and the remaining drive mechanisms, or the vibration in the test rig. The circumferential force was observed to significantly rise together with increasing revolutions. Interestingly, standard deviation decreases together with rising motor revolutions (for the upper ranges of belt linear speeds).

### 2.2. Sensors Options

Tests of drive mechanisms in contact require performing high-accuracy measurements of belt linear speed. The originally proposed measurement system comprising a pair of incremental encoders located at the beginning and at the end of the frictional contact length in the intermediate drive, each having an accuracy of up to 0.01 m/s, proved insufficient at recording the slippage difference between the belts in contact. This problem was due to the fact that the frictional contact length of 5 m was not sufficiently long and the elastic deformations of both belts were limited. Therefore, an attempt was made to search for a different measurement technique, which would ensure more accurate belt speed identification and which would allow future tests of the influence of slippage between the tension members on the transmitted friction force.

The literature indicates numerous examples of belt linear speed measurement methods. The usefulness of these various techniques is determined mostly by the accuracy adjusted to the investigated physical phenomena and by the technical potential for adapting the measurement system to the conditions of a particular experiment (laboratory and in-service tests). The photoelectric incremental encoders [3] used in the preliminary research, along with self-calibratable encoders [4] or easily miniaturizable optical encoders [5], belong to a popular group of sensors allowing belt speed to be continuously recorded with the use of the contact method. They are successfully employed in monitoring the operation of, e.g., belt transmission systems [6] or evaluating in small-scale the slippage and transverse vibration in belt drives [7,8]. Despite their high measurement accuracy declared by the manufacturers to be ≤0.01 m/s, a note should be made that the unsteady motion of the belt (lateral wandering) and transverse vibration which occur during the continuous recording of belt speed may disturb the signal and lead to permanent changes in the geometry of the rotators [9].

Another group of sensors used in speed measurements is based on contactless optical methods. One such interesting solution is, for example, sensors measuring reluctance variability. They use magnetoresistivity, in which an external magnetic field induces a change in the resistivity of the material [10]. The problems related to the contact between the measuring apparatus and the conveyor belt can be also eliminated by using inductive sensors. They prove particularly useful in operating conditions, in addition to being reliable and having quick reaction time when the object is introduced into the sensor active zone [11,12]. Another speed measurement method may be based on a measurement system comprising an infrared (IR) transmitter and receiver (infrared sensor) which uses the phenomenon of light absorption [13]. In this case, black and white markers (discs) are attached to the moving element. Their white surfaces reflect all of the infrared radiation, while their black surfaces absorb it. Moreover, electrostatic sensors can be used, which allow belt speed to be identified on the basis of the records of belt vibration levels [14]. Tests performed with belt linear speed within the range of 2–10 m/s demonstrated that this type of measurement is subject to ± 2% relative error. 

Publication [9] presents the results of speed tests with the use of a CCD camera pointed at the bottom side of the conveyor belt marked with special measurement marks. During the tests, the camera was switched to a high-speed multi-series photography mode. With a number of photos taken over a short time, changes of the measurement fields defined by the markers could be observed during the movement of the belt. Based on the known number of photos taken at a constant time interval and on the shifts of the characteristic points in the direction of belt movement, a measuring algorithm calculated belt speeds in real time. Calculations also covered Mean Absolute Error (MAE) = 0.029 m/s and Root Mean Square Error (RMSE) = 0.042 m/s. Similar tests were also performed for a belt conveyor on a laboratory scale, with the use of a CCD camera, following belt edges. The obtained RMSE was 0.018 m/s, and MAE was 0.010 m/s, with tests in the range of 0 ÷ 3 m/s [15].

## 3. Materials and Methods

Based on the preliminary pilot tests and on the literature review, three belt linear speed measurement methods were selected for tests on the test rig. The methods employed were:Incremental encodersInductive sensorsHigh-speed camera

The tests started from the previously used incremental sensors, in order to compare the usefulness of the new methods in identical experiment conditions. The selection of the two new methods was based on the estimated accuracy level and their usefulness in laboratory tests.

In each of the three methods, measurements of belt linear speed were performed in the entry zone of the belt on the head pulley of the intermediate drive. Figure 3 schematically shows the positions of the sensors and the camera with respect to the test rig. The tests were performed at 19.5 °C, in the entire available range of belt linear speeds, i.e., from 1.40 to 4.15 m/s. The speeds were changed on six levels, corresponding to the following revolutions on the intermediate drive motor: 500, 700, 900, 1100, 1300 and 1500 rpm.

### 3.1. Incremental Encoder

Measurements with the use of an incremental encoder require a measurement wheel remaining in constant contact with the moving belt. The wheels used in the test were covered with a special cover to ensure stable operation without slippage relative to the belt. In addition, the sensor was mounted in a holder with a broad range of regulated positions, which allowed it to be clamped to the belt with a constant force. The encoder used in the test could record 2000 pulses per one wheel revolution (Figure 4). The known number of pulses per one revolution and the circumference of the measurement wheel allowed calculations of belt linear speed. The recorded series of belt linear speed for the set rotational speed of the motor are shown in Figure 5. 

### 3.2. Inductive Sensor

For the purposes of the test, a measurement system was constructed from an inductive sensor and a microcontroller *AVR ATmega328P-U DIP*, which sent the measured data to the monitor of the serial port in the computer. The system was powered by direct current of 5 V for the microcontroller and of 12 V for the inductive sensor. The inductive sensor sent logical values, i.e., a zero-one signal. It facilitated the communication between the inductive sensor and the microcontroller with the use of the opto-isolator PC 817, via one of the available digital channels. The test was performed with special steel markers attached to the belt. The inductive sensor was fastened at the entry point of the belt to the head pulley of the intermediate drive (Figure 6), where the force in the belt is the greatest and the vibration occurs in a relatively limited range. Markers of an adequate width additionally reduced the impact of belt lateral wandering on the test results. Figure 7b,c is a schematic diagram of the test rig.

At the moment when the marker occurred in the detection zone, the inductive sensor sent a pulse via the opto-isolator PC 817 to the digital channel of the microcontroller *AVR ATmega328P-U DIP.*
Figure 7a is the schematic diagram of the connections. Calculations of the belt linear speed were possible with the known distances between the successive markers and the measured times between pulses generated by the inductive sensor. 

Figure 8 shows the calculated belt linear speed series recorded during the tests with the use of the inductive sensor.

### 3.3. High-Speed Camera

The tests were performed with the use of the Phantom v2640 camera (Ametek Materials Analysis Division, USA) for time lapse recording with a 4 Mpx matrix (Figure 9). The recorded images, which served to calculate vectors of instantaneous speeds for each of the measurement points, were analyzed with the Phantom Camera Control (PCC) software. It enabled the motion in the recorded images to be analyzed on the basis of the calculated distances, angles, linear and angular speeds of the moving objects with a maximum number of 4 measurement points. This, in turn, provided the possibility to calculate relative differences between the movement parameters of the objects, as well as absolute values, with the adequate calibration of the image from the camera. The image was calibrated with a measuring rod of known graduation, located horizontally over the conveyor belt (Figure 10).

The camera was positioned at a distance of 4.35 m from the belt conveyor, perpendicular to the belt movement. The camera lens was pointed at the measurement length of 0.1 m. Measurement markers of known dimensions were located on the edge of the moving conveyor belt (Figure 10). The centers of these markers were located at the contact of four contrasting fields and were assumed as the measuring points for determining vectors of instantaneous belt speeds. The known dimension of the marker enabled future calibrations of the obtained image and calculations of absolute belt speed.

The belt in the observation range of the camera moves not only horizontally. The horizontal movement dominates, but the sag of the belt and the vibrations due to the operation of the rig also cause vertical displacements. In order to analyze the recorded belt movement, a comparison was made of the displacement areas of the markers in their vertical plane of movement (Figure 11). Horizontal displacements dominate in the direction of belt movement and displacements in the vertical axis Y are negligible (smaller by an order of magnitude). The recorded vertical displacements neither reflect the theoretical belt sag curve nor show the properties of belt transverse vibration. They can be assumed to be random disturbances in the measurements. Therefore, a decision was made to use only horizontal displacements in the X axis as a basis for calculating belt linear speed.

The location of the markers in the central part of the core, on the belt edge, additionally ensured that the direction of belt movement and its speed relative to the camera were precisely reflected [16]. The instantaneous speed of an observed point on the belt edge is defined by the following relationship:(1)$$\begin{array}{c} v=\frac{\Delta s}{\Delta t}=\frac{s\;\cdot \;F}{k_{n}-k_{n-i}}\; \end{array}$$
where:Δs—point displacement, Δt—point displacement time,s—actual distance between the characteristic sections in the image from the camera,F—number of frames recorded per second, and$k_{n}$, $k_{n-i}$—numbers of frames in which the measured point was located between the characteristic sections in the image. 

Figure 12 shows the linear speeds calculated for six measurement series. For each of the analyzed rpm ranges, the analysis covered belt speed in five measuring points applied to the belt edge at intervals 0.1 m in length. 

## 4. Results

The results for each of the methods were objectively analyzed for accuracy with the use of two error measures: mean absolute error (MAE) and root mean square error (RMSE) [17]. The values of these indicators tending to zero indicate the high accuracy of the measurement method. 

The measurement errors calculated for each of the methods with the use of the above equations showed an increasing dependence on the motor revolutions, which allowed approximation with the linear functions shown in Figure 13.

The functions of the MAE/RMSE error values and the motor revolutions for each of the measurement methods can be described as follows (2):(2)MAE / RMSE=a · RPM+b 

Table 1 contains the matching parameters of these functions described with Equation (2) for the corresponding error values, together with the coefficient of determination R^2^.

The highest accuracy of the three tested methods was obtained with the high-speed camera Phantom v2640 for the belt speed in the range of 1.42 to 4.15 m/s. The accuracy indicators for this method are at MAE = 0.0088 m/s and RMSE = 0.0112 m/s, and are significantly higher than in the case of the inductive sensor method, in which the measurement errors are the highest: MAE = 0.0514 m/s, RMSE = 0.0660 m/s. The inductive sensor method is also influenced to the greatest extent by the measurement-disturbing factors. 

Table 2 compares the accuracies of own measurements with the accuracies obtained in other research centers [8,13,14,17]. This comparison proves that high-speed camera methods provide much higher accuracies than the traditional methods based on contact or inductive sensors. 

## 5. Discussion

The test results demonstrated that the high-speed camera method is useful in measuring belt speed with accuracy, enabling observations of elastic slippage in the frictional contact lengths. The method was further verified in research performing measurements of the belt on the wrap arc of the head drive pulley in the main conveyor. The method consists of calculating mean speeds in selected sections along the belt movement path. In this case, eight identical sections were selected on the pulley wrap arc (Figure 14). The total pulley wrap arc length of the belt was approximately 1 m, and the length of each section was 0.12 m. The obtained results are shown in Figure 14. The line graph shows linear belt velocity with blue line, measured by the high-speed camera. The red line represents the tangential velocity of the drive pulley during the measure series. The character of changes in the measured speed is consistent with prior analyses of belt frictional contact on the drive pulley with consideration of the shear deformation of the belt bottom cover and the pulley rubber coating, as well as to the longitudinal strain (contraction) of the belt core [19]. The analyses indicate that the initial area in which belt speed increases comprises a zone where tangential stresses on the contact surface between the bottom cover and the pulley rubber coating do not exceed the friction limit condition—this is the rest zone. Frictional contact continues in the elastic slippage zone—an area in which belt speed decreases. The ratio between these two zones should depend on the magnitude of the circumferential force transmitted on the pulley. Analysing Figure 14, a characteristic adhesion and slip areas were observed. This is evidenced by the increase in linear belt speed to the limit point and the decrease. The distribution of speed on the pulley is associated with a decrease in the tensile force in the belt on the friction contact arc. Equally important is the shear deformation of the rubber in the contact zone caused by the bending of the belt on the pulley. The obtained force in the belt did not allow us to record a significant difference in speed between the initial and the last point of tangency of the belt with the pulley.

## 6. Conclusions

The tests were performed in constant laboratory conditions which ensured an objective comparison between the results obtained for the three tested methods. Belt speed was measured in the belt entry zone on the head drive pulley of the intermediate drive system, where both belts move linearly and vibration occurs in a relatively limited range. 

The usefulness of the three test methods was evaluated with standard measures, i.e., mean absolute error (MAE) and root mean square error (RMSE). For each of the three methods, the measures were directly proportional to the set belt speed and show a linear relationship.

The highest accuracy of belt linear speed measurements was obtained with the use of the high-speed camera method. In the case of tests performed on the test rig within the belt speed range of 1.42–4.15 m/s, this method allows very high data-reading frequency and limits the impact of belt vibration on the measurement results. The inductive sensor method proves to be the least useful, not only because it provided results with the highest measurement errors, but also because the method itself demonstrated its limited tolerance to dynamic factors which negatively influence the measurement results. 

During the tests with the incremental encoder, at time-varying torque, the set reading frequency was observed to influence the accuracy of belt speed measurements. The sensor holder should be importantly equipped with a compression spring, in order to ensure constant contact between the measurement wheel and the belt during the tests, as a temporary loss of contact (e.g., due to vibrations) significantly influences the quality of the results.

The inductive sensor used in the tests, which had a measurement range of approximately 5 mm, showed a very limited tolerance to the vibrations of the test rig. As a result, the marker frequently left the measurement range of the sensor. Importantly, in the case of this method, the reading frequency depends on the number of markers placed on the belt, as well as on the length of the belt loop. 

In high-speed camera tests, the measuring point moves in the vertical plane with two speed components (horizontal and vertical). Therefore, on each occasion of defining the experiment conditions, attention should be paid to evaluating the influence of the vertical displacements on the horizontally measured linear speed of the point.

Based on laboratory tests of measurement methods adapted to determine the linear speed of the belt, their usefulness was assessed in the elastic slip test. Slippage tests were carried out under laboratory conditions, so the selection criterion is the measurement accuracy. The high-speed camera produced the highest accuracy results (MAE = 0.0088 m/s, RMSE 0.0112 m/s), making it the best measurement method. The encoder (MAE = 0.0171 m/s, RMSE 0.0215 m/s) and the inductive sensor (MAE = 0.0514 m/s, RMSE 0.0660 m/s) showed much lower accuracy.

The results of high-speed camera tests require highly labor-intensive processing and therefore efforts should be made to automate the procedures in order to ensure repeatability of the performed operations and to accelerate the results.

The example of belt speed measurement on the drive pulley demonstrated the usefulness of the optical method based on a high-speed camera in tests of elastic slippage in the frictional contact section.

## Figures and Tables

**Figure 1 sensors-20-05816-f001:**
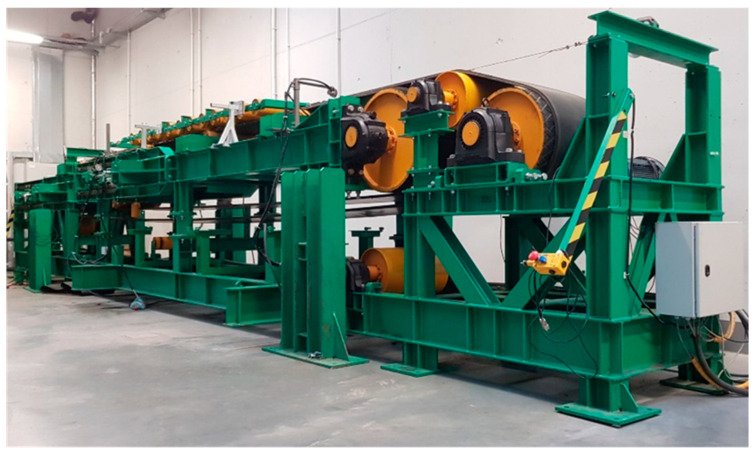
Intermediate belt drive test rig.

**Figure 2 sensors-20-05816-f002:**
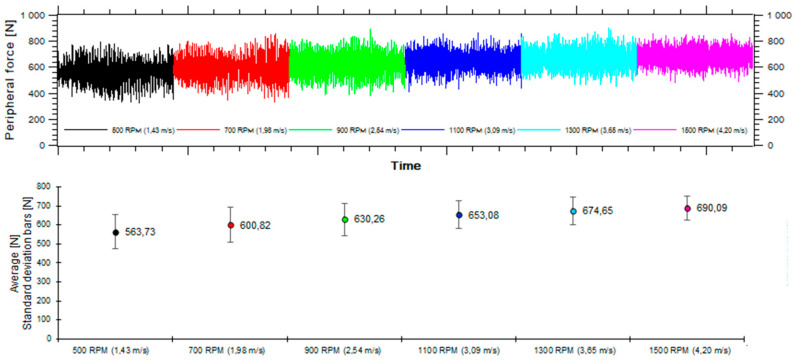
Variability of circumferential force recorded in the full range of belt speeds, at constant belt tension and load.

**Figure 3 sensors-20-05816-f003:**
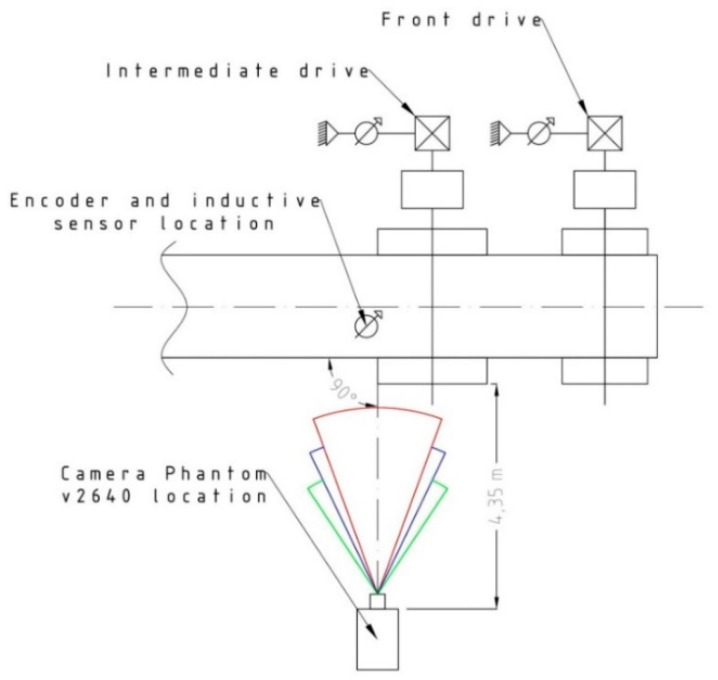
Positions of belt linear speed measurements on the intermediate drive test rig.

**Figure 4 sensors-20-05816-f004:**
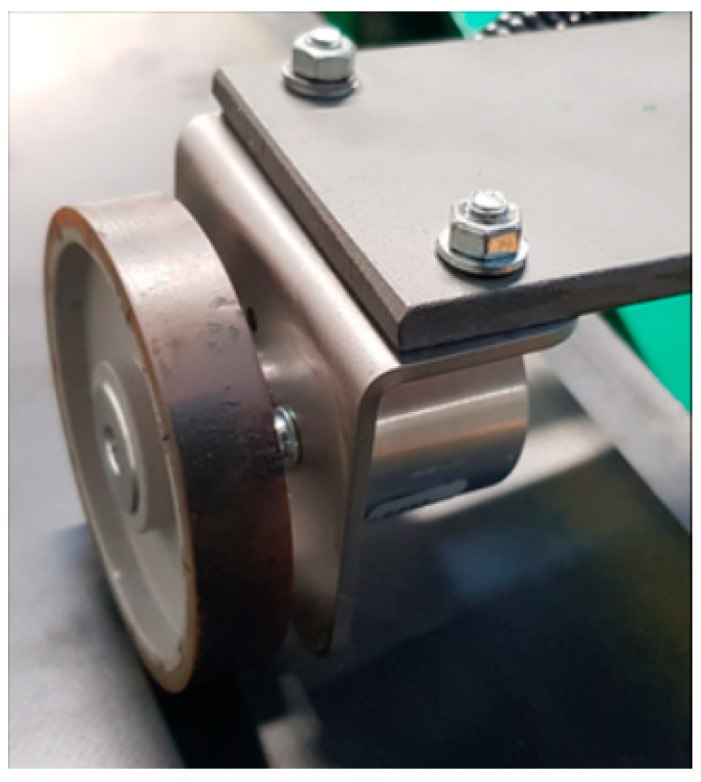
View of the incremental encoder in contact with the belt in the test area.

**Figure 5 sensors-20-05816-f005:**
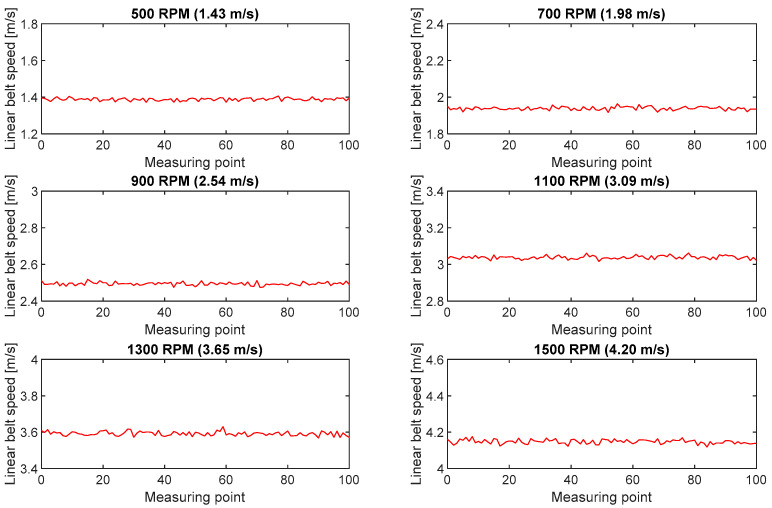
Series of belt linear speeds calculated in the measurements with the use of the incremental encoder for six rpm ranges of the pulley motor.

**Figure 6 sensors-20-05816-f006:**
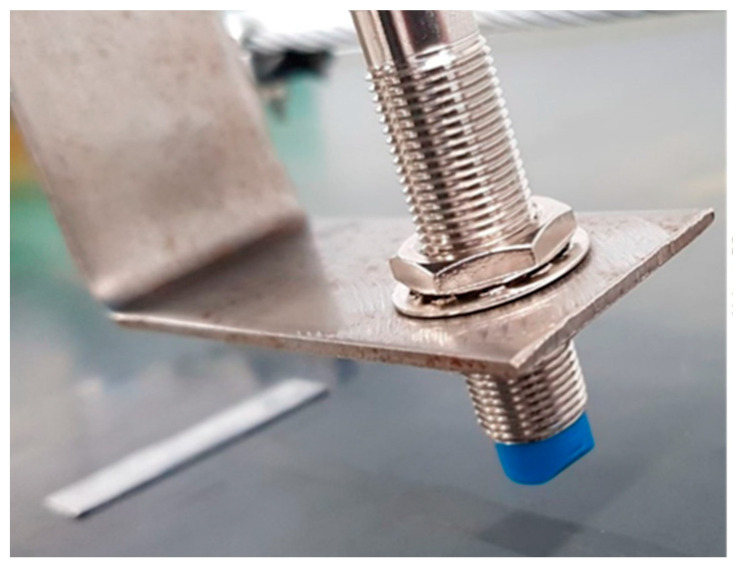
Inductive sensor at the measuring point.

**Figure 7 sensors-20-05816-f007:**
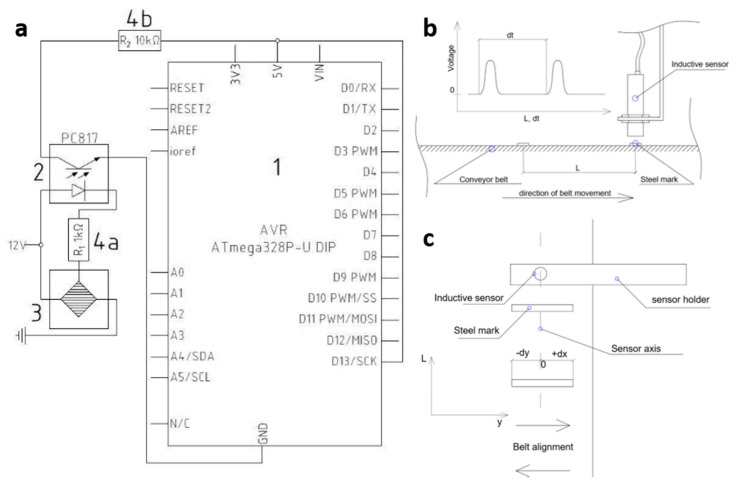
Measurement with the inductive sensor: (**a**) conveyor belt linear speed measurement system with inductive sensor (1—microcontroller, 2—linear opto-isolator, 3—inductive sensor, 4—resistors), (**b**) measurement with the inductive sensor—perspective view, (**c**) measurement with the inductive sensor—top view.

**Figure 8 sensors-20-05816-f008:**
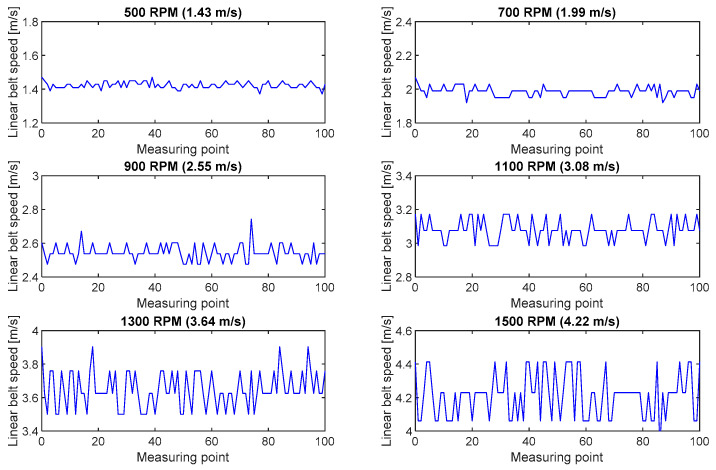
Fluctuations of belt linear speeds calculated in the inductive sensor measurements for six rpm ranges of the pulley motor.

**Figure 9 sensors-20-05816-f009:**
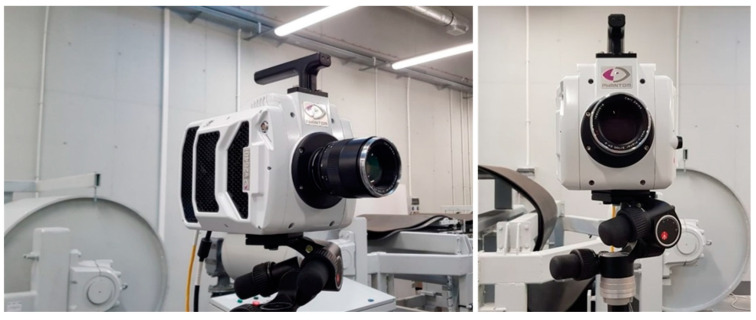
View of the Phantom v2640 camera at the measurement location.

**Figure 10 sensors-20-05816-f010:**
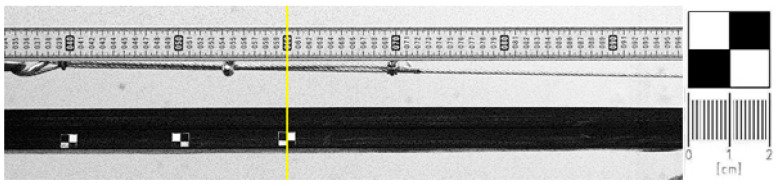
Image calibration—view of the belt with the measuring points and dimensions of the markers.

**Figure 11 sensors-20-05816-f011:**
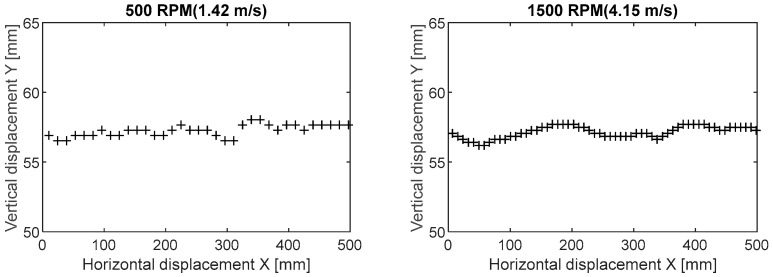
Vertical displacement of the measurement point as a function of horizontal displacement (distance traveled).

**Figure 12 sensors-20-05816-f012:**
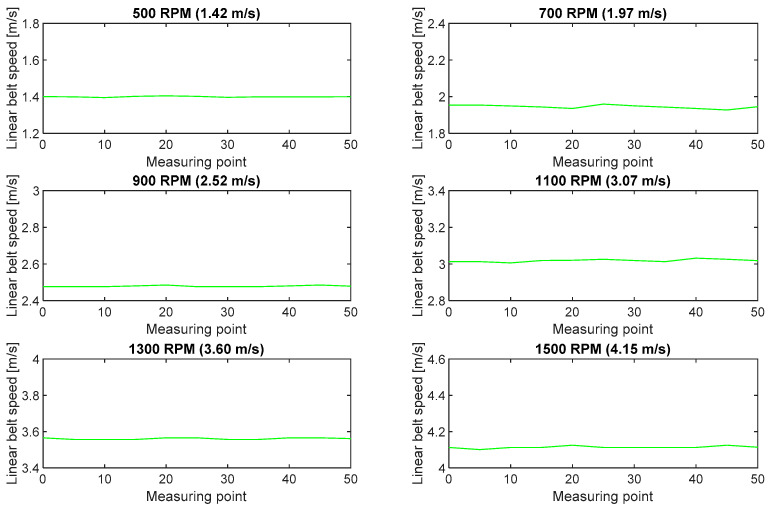
Fluctuations of belt linear speeds calculated in the measurements with the use of the high-speed camera for six rpm ranges of the pulley motor.

**Figure 13 sensors-20-05816-f013:**
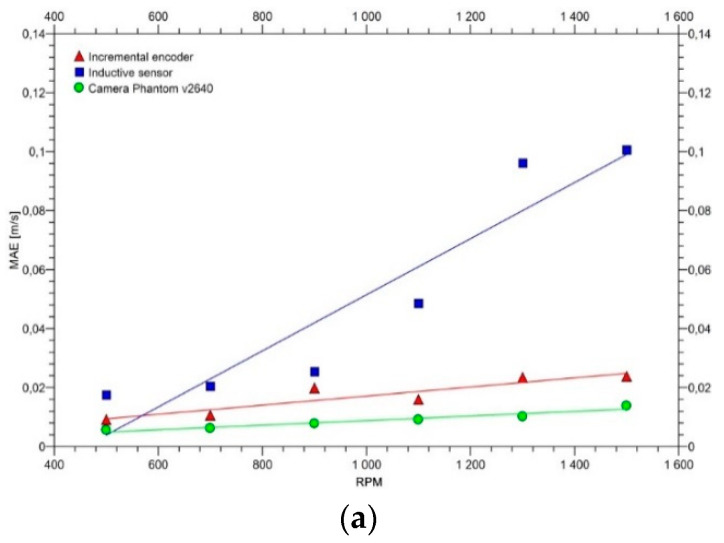
Errors calculated for the three selected measurement methods in the full range of the recorded revolutions per minute: (**a**) MAE, (**b**) RMSE.

**Figure 14 sensors-20-05816-f014:**
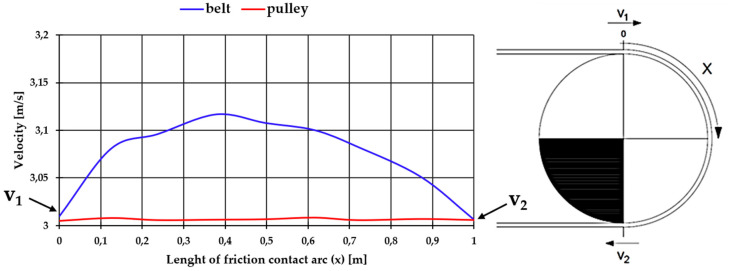
Belt velocity distribution on the length of friction contact arc of the drive pulley and pulley surface velocity for one case (average linear belt velocity 3.07 m/s, belt pre-tension 33 kN, torque on the pulley shaft 400 Nm, belt surface temperature 292.65 K).

**Table 1 sensors-20-05816-t001:** Matching parameters of linear functions described with Equation (2).

***RMSE = f(RPM)***			
**Method**	**a**	**b**	**R^2^**
**Incremental encoder**	0.000017	0.004084	0.930652
**Inductive sensor**	0.000117	−0.050964	0.963096
**Camera Phantom v2640**	0.001488	0.001117	0.943409
***MAE = f(RPM)***			
**Method**	**a**	**b**	**R^2^**
**Incremental encoder**	0.000015	0.001673	0.913642
**Inductive sensor**	0.000095	0.043651	0.936934
**Camera Phantom v2640**	0.000008	0.001005	0.966184

**Table 2 sensors-20-05816-t002:** Compared accuracies of selected linear speed measurement methods; X—no data available.

Method	Relative Error [%]	MAE [m/s]	RMSE [m/s]
**Incremental Encoder**	0.03%	0.0171	0.0215
**Inductive Sensor**	0.30%	0.0514	0.0660
**Camera Phantom v2640**	0.06%	0.0088	0.0112
**CCD Camera** [15]	X	0.0100	0.0180
**CCD Camera** [9]	X	0.0290	0.0420
**Camera** [18]	2.00%	X	X
**Electrostatic sensor** [14]	2.00%	X	X

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
