# Peer review of "Tests of Belt Linear Speed for Identification of Frictional Contact Phenomena"

_sensors, 2020, doi:10.3390/s20205816_

Round 1

Reviewer 1 Report

Reviewer’s comments on the manuscript “Tests of Belt Linear Speed for Identification of Frictional Contact Phenomena”, submitted by Piotr Bortnowski, Lech GÅ‚adysiewicz, Robert Król and Maksymilian Ozdoba for possible publication in Sensors

  • In the Introduction it is not clear, what is the “elastic slippage” and what is the “sliding slippage”. Is the “elastic slippage” the relative motion between the belt and the pulley surface because of the rubber coating and thus the shear deformability of the pulley surface?
  • Beginning of section 2 – that is probably an excerpt from the Instructions for the Authors, not the intended text of the paper.
  • Do vertical vibrations of the belt affect the position and the size of the sliding zones, making them thus rapidly fluctuating in time?
  • The term “non-dilatiational strain”, used in section 4, is not familiar to me and probably needs additional explanation in the text.
  • The optical sensor measures the velocity of the outer belt surface. The inner surface, which is actually in contact with the pulley surface, may have different velocity because of the two effects:
    • The curvature of the belt times its thickness determines the relative velocity difference because of the obvious geometric effect – how high is this parameter for your setup? What is the relation of the exactly known pulley surface velocity to the measured belt speeds for the experiment in Fig. 14?
    • The shear deformability of the belt may be as important as its elastic or sliding slippage, see
      Alciatore, D. G., and A. E. Traver, 1995. "Multipulley belt drive mechanics: creep theory vs shear theory.", J.Mech.Des. 117(4), 506-511
  • When discussing the experiment presented in Fig. 14, you say that v1 is significantly higher than v2, which is not really visible in the plot – the velocity variations inside the contact domain are much more significant.
  • In section 4 one reads: “Frictional contact continues in the elastic slippage zone – an area in which belt speed decreases” – that is more than the half of the contact arc in the presented plot. Earlier it is frequently said, that the friction contact lengths are negligibly short in comparison to the total length, which poses a certain contradiction.
  • A short theoretical overview of the mechanical behavior of the moving belt with zones of sliding friction would probably be useful for the reader. Additionally to the above reference, the discussion may rest upon the following ones:
    Scheidl, J. and Vetyukov, Y., 2020. Steady motion of a slack belt drive: dynamics of a beam in frictional contact with rotating pulleys. Journal of Applied Mechanics, in press, pp.1-42
    Rubin, M.B., 2000. An exact solution for steady motion of an extensible belt in multipulley belt drive systems.  Mech. Des.122(3), pp. 311-316.

Author Response

Dear Reviewers

Thank you for the useful comments that helped to improve the quality of our manuscript. Below we address each of the reviewer’s comments and our responses are marked italics.

Of course, all changes have been added in the article. Our changes are marked red.

Reviewer 2 Report

I have reviewed the paper based on what the authors set out to achieve.

1. Obtaining good test results from working test rigs is always a daunting task. The author's measurements and tests of 3 types of sensors show the optical method has the cleanest electronic processed signal. So I have no real technical disagreements or negative comments on the content of the paper, due to what the authors set out to present.

2. Relative to the above comments and objectives, I can't be critical. But I can say that the paper is lacking theoretical content. I accept the paper is in a Sensor Journal, not a Theoretical Mechanical Engineering Journal. The paper is therefore considered more experimental in nature.

3. The authors should, nevertheless be exposed to expert thinking on this subject, this will also help make the Journal paper stronger and more encompassing. In an engineering research context, the paper does not address more academic aspects of the problem. For example, in the "Discussion Section" a true researcher who is knowledgeable in the field would address and "Reference" the following points: -

a) Real-world issues related to reliability and usage of optical devices in dusty and humid environments, such as conveyors in mines.

b) A theoretical discussion of test rig issues, i.e. low tension differential around drive drums in a simple test rig like used in the laboratory. Specifically, the difference in belt tension entering (T1) and exiting (T2) a real conveyor drive might be 50 to 100 times greater than achievable on a lab test rig, therefore the elastic stretch, and hence higher velocity differences that would be measured, are highly significant in terms of slip and wear.
References :
Harrison, A.: Modelling Belt Tension around a Drive Drum. Bulk Solids Handling Int. J. Vol. 18, No. 1, pp 75-80 (1998)
c) The effect of drum lagging on speed differential. In the authors paper, the magnetic method may be more practical due to the environment. Magnetic devices may also give much larger output differentials idf the T1 - T2 is much greater, as is the case for real conveyors.

d) No graphs of belt tension or stretch / relaxation entering and exiting the drive drums of the test rig are given. So for other researchers the small speed differences measured are not scalable to bigger systems without such graphs.

Editorial Organisational Changes are Required.

a) There are Duplicate Section 2 - Materials and Methods.

b)  There is no Conclusion Section !

c) The Introduction is too long and contains findings normally placed in later sections.  Change as follows :  Break the Introduction at the top of page 2 starting with "Tests ..........

So, do the following format changes :

  1. Introduction (Page 1)  Shorten and add a new section as follows.
  2. Procedures (Page 2) - New Sub-Section 2.1 Laboratory Facility (Page 2) commencing at "Test .....  and then New Sub-Section 2.2 Sensor Options (Page 4) and Delete First Duplicated Heading  2. Materials and Methods.
  3. Materials and Methods  (Page 5)     3.1    3.2   3.3  are as is, OK.
  4. Results and Discussion
  5. Conclusion

Technically the paper is acceptable.

After these changes are adopted I could recommend publication, after I see that the changes were correctly understood and recommended.

Author Response

(The authors gave the same response as above.)

Reviewer 3 Report

The paper presents an evaluation of three methods for belt speed identification.

It is not clear the paper goals and contributions. Are the authors introducing new methods or improved methods? Are they presenting an evaluation of the knowledge methods of the literature?

Also, the paper lacks some details for the best results evaluation.

There are two sections named "Materials and Methods"

What do mean parameters 'a' and 'b' of Equation 4 and Table 1. How were they found?

Round 2

Reviewer 2 Report

none

Author Response

Thank you

Reviewer 3 Report

Thanks for the answer and changes in the paper content. 

I understood the contributions, but my last suggestion is to include in the introduction a sentence similar to:

"The paper contributions are:

  • Contribution A
  • Contribution B
  • Contribution C"

I believe this can increase the interest in reading of your paper. 
